# DIFFUSION IN DIFFUSION: CYCLIC ONE-WAY DIFFUSION FOR TEXT-VISION-CONDITIONED GENERATION

**Ruoyu Wang**[1*]**, Yongqi Yang**[1*]**, Zhihao Qian**[1]**, Ye Zhu**[2]**, Yu Wu**[1†]
[1] School of Computer Science, Wuhan University
[2] Department of Computer Science, Princeton University
{wangruoyu, yongqiyang, qianzhihao, wuyucs}@whu.edu.cn
yezhu@princeton.edu

## ABSTRACT

Originating from the diffusion phenomenon in physics that describes particle movement, the diffusion generative models inherit the characteristics of stochastic random walk in the data space along the denoising trajectory. However, the intrinsic mutual interference among image regions contradicts the need for practical downstream application scenarios where the preservation of low-level pixel information from given conditioning is desired (e.g., customization tasks like personalized generation and inpainting based on a user-provided single image). In this work, we investigate the *diffusion (physics) in diffusion (machine learning)* properties and propose our *Cyclic One-Way Diffusion (COW)* method to control the direction of diffusion phenomenon given a *pre-trained frozen diffusion model* for versatile customization application scenarios, where the low-level pixel information from the conditioning needs to be preserved. Notably, unlike most current methods that incorporate additional conditions by fine-tuning the base text-to-image diffusion model or learning auxiliary networks, our method provides a novel perspective to understand the task needs and is applicable to a wider range of customization scenarios **in a learning-free manner**. Extensive experiment results show that our proposed *COW* can achieve more flexible customization based on strict visual conditions in different application settings. Project page: https://wangruoyu02.github.io/cow.github.io/

## 1 INTRODUCTION

In physics, the diffusion phenomenon describes the movement of particles from an area of higher concentration to a lower concentration area till an equilibrium is reached (Philibert, 2006). It represents a stochastic random walk of molecules to explore the space, from which originates the state-of-the-art diffusion generative models (Sohl-Dickstein et al., 2015). Compared to the physical diffusion process, it is widely acknowledged that the diffusion generative model in machine learning also stimulates a random walk in the data space (Song et al., 2020b), however, it is less obvious how the diffusion models stimulate the information diffusion for real-world data along its walking trajectory. In this work, we start by investigating the diffusion phenomenon in diffusion models for image synthesis, namely *"diffusion in diffusion"*, during which the pixels within a single image from different data distributions exchange and interact with each other, ultimately achieving a harmonious state in the data space (see Sec. 3.2).

The diffusion phenomenon is intrinsically stochastic, both in physics and in current machine learning frameworks, given no explicit constraints on the regional concentrations. In other words, regions within an image interfere with each other along the generation process as the sample goes from a noisy Gaussian space to the data space. However, we note that this property of bidirectional interference is not always desired when applying diffusion generative models in practical downstream tasks. For instance, tasks like image inpainting can be viewed as strictly unidirectional information diffusion, propagating information from a known portion (the existing image portion)

---

*Equal contribution.
†Corresponding author.

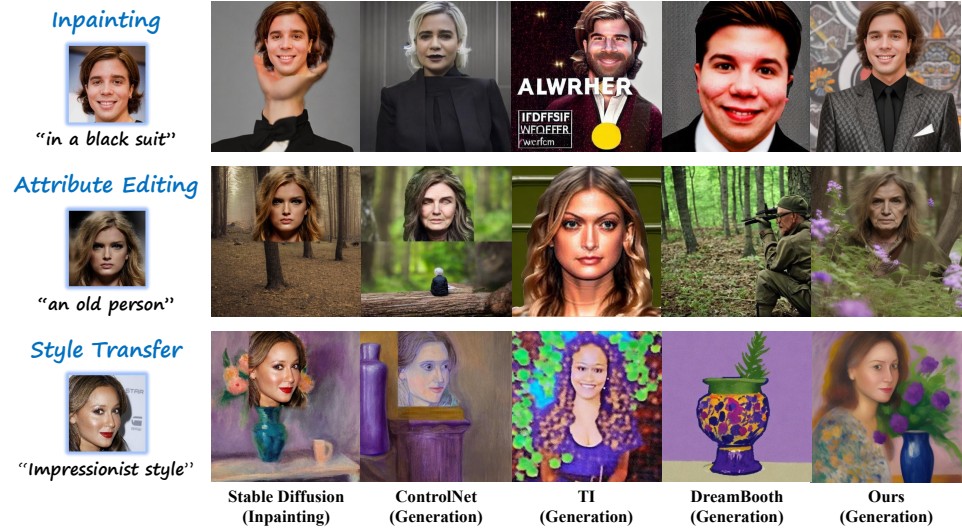

Figure 1: Comparison with existing SOTA methods for maintaining the fidelity of text and visual conditions in different application scenarios. We consistently achieve superior fidelity to both text and visual conditions across all three settings. In contrast, other learning-based approaches struggle to attain the same level of performance across diverse scenarios.

to an unknown portion (the missing image portion) while preserving the pixel-level integrity of the known portion. Most existing methods (Ruiz et al., 2022; Gal et al., 2022; Dong et al., 2022; Zhang & Agrawala, 2023) tackle the problem by brute-force learning, where they incorporate the visual condition into pre-trained text2image models by introducing an additional finetuning process to minimize the reconstruction error. However, these tuning-based methods, despite explicitly minimizing reconstruction errors, can not always achieve pleasant fidelity to both the text and the visual conditions, especially when faced with contradictions between conditions like style transfer and mismatched attributes scenarios, as shown in Fig. 1. In addition, they introduce additional learning costs dependent on the pre-trained model and hinder the original distribution modeling ability of the base model. To this end, the ability to control the direction of information diffusion opens up the potential for a new branch of methodological paradigms to achieve versatile customization applications *without the need to change the parameters of existing pre-trained diffusion models or learn any auxiliary neural networks*.

Following our analysis on *the diffusion in diffusion* and its connection to practical task needs, we propose *Cyclic One-Way Diffusion* (COW), a training-free framework that achieves unidirectional diffusion for versatile customization scenarios, ranging from conventional visual-conditioned inpainting to visual-text-conditioned style transformation. From the methodological point of view, we re-inject the semantics (inverted latents) into the generation process and repeatedly "disturb" and "reconstruct" the image in a cyclic way to maximize and encourage information flow from the visual condition to the whole image. From the application point of view, the powerful knowledge of the pre-trained diffusion model empowers us to conduct meaningful editing or stylizing operations while maintaining the fidelity of the visual condition. As a result, even in its unidirectional design, COW can achieve more flexible customization based on strict visual conditions. Extensive experiments and human studies, involving 3,000 responses for 600 image groups, demonstrate that COW consistently outperforms its counterparts in terms of condition consistency and overall fidelity. Besides, COW generates an image in just 6 seconds, far faster than other customization methods like DreamBooth (Ruiz et al., 2022), which takes 732 seconds.

## 2 RELATED WORK

**Diffusion Models.** The recent state-of-the-art diffusion generative methods (Song et al., 2020a; Dhariwal & Nichol, 2021; Nichol & Dhariwal, 2021; Song et al., 2020b; Hoogeboom et al., 2021; Wu et al., 2022; Zhu et al., 2023b) originate from the non-equilibrium statistical physics (Sohl-

Dickstein et al., 2015), which simulate the physical diffusion process by slowly destroying the data structure through an iterative forward diffusion process and restore it by a reverse annealed sampling process. DDPM (Ho et al., 2020) shows the connection between stochastic gradient Langevin dynamics and denoising diffusion probabilistic models. DDIM (Song et al., 2020a) generalizes the Markovian diffusion process to a non-Markovian diffusion process, and rewrites DDPM in an ODE form, introducing a method to inverse raw data into latent space with low information loss. In our work, we utilize DDIM inversion to access the latent space and find an information diffusion phenomenon in the sampling process towards the higher concentration data manifold driven by the pre-trained gradient estimator.

**Downstream Customization Generation.** Given a few images of a specific subject, the customization generation task aims to generate new images according to the text descriptions while keeping the subject's identity unchanged. Early approaches mainly relied on GAN-based architectures (Reed et al., 2016; Zhang et al., 2017; Dong et al., 2017; Zhang et al., 2018; Xu et al., 2018; Lin et al., 2018; Wu et al., 2019; Karras et al., 2019; Li et al., 2019; Karras et al., 2020) for customization generation. In recent years, the diffusion methods under text condition image generation task (T2I) have made a great development (Nichol et al., 2021; Ramesh et al., 2022; Oppenlaender, 2022; Ramesh et al., 2021; Ho & Salimans, 2022; Saharia et al., 2022). However, appointing a specific visual condition at a certain location on the results of a generation remains to be further explored. Choi et al. (2021) is a learning-free visual conditioned method that generates a random new face similar to the given face. There are recent customized methods of learning-based visual-text conditioned generation like Ruiz et al. (2022); Gal et al. (2022); Zhang & Agrawala (2023); Rombach et al. (2022); Dong et al. (2022); Brooks et al. (2023); Wu et al. (2023); Wei et al. (2023); Guo et al. (2023); Choi et al. (2021). Methods like DreamBooth (Ruiz et al., 2022) and Textual Inversion (Gal et al., 2022) learn the concept of the visual condition into a certain word embedding by additional training on the pre-trained T2I model (Kumari et al., 2022; Dong et al., 2022; Ruiz et al., 2023; Chen et al., 2023). It is worth noting that it is still hard for DreamBooth or Textual Inversion (TI) to keep the id-preservation even given 5 images as discussed in Shi et al. (2023), and DreamBooth tends to overfit the limited fine-tuning data and incorrectly entangles object identity and spatial information, as discussed in Li et al. (2023). ControlNet (Zhang & Agrawala, 2023) trains an additional network for a specific kind of visual condition (e.g., canny edges, pose, and segmentation map). More generally, an in-painting task is also a customization generation with a forced strong visual condition. Compared to these methods, our proposed method can explicitly preserve the pixel-level information of the visual conditions while achieving versatile application scenarios like style transfer and attribute editing.

## 3 METHOD

In this section, we present our methodology design for leveraging the diffusion in diffusion phenomenon to achieve versatile downstream applications without additional learning.

### 3.1 PRELIMINARIES

Denoising Diffusion Probabilistic Models (DDPMs) (Ho et al., 2020) define a Markov chain to model the stochastic random walk between the noisy Gaussian space and the data space, with the diffusion direction written as,

$$q(\mathbf{x}_t|\mathbf{x}_{t-1}) = \mathcal{N}(\sqrt{1-\beta_t}\mathbf{x}_{t-1}, \beta_t I), \tag{1}$$

where $t$ represents diffusion step, $\{\beta_t\}_t^T$ are usually scheduled variance, and $\mathcal{N}$ represents Gaussian distribution. Then, a special property brought by Eq. 1 is that:

$$q(\mathbf{x}_t|\mathbf{x}_{t-k}) = \mathcal{N}(\sqrt{\alpha_t/\alpha_{t-k}}\mathbf{x}_{t-k}, \sqrt{1-\alpha_t/\alpha_{t-k}}I), \tag{2}$$

where $\alpha_t = \prod_{i=0}^{t}(1-\beta_i)$. So we can bring $\mathbf{x_t}$ to any $\mathbf{x}_{t+k}$ in a one-step non-Markov way in our proposed cyclic one-way diffusion (Sec. 3.3) by adding certain random Gaussian noise.

DDIM (Song et al., 2020a) generalizes DDPM to a non-Markovian diffusion process and connects the sampling process to the neural ODE:

$$d\overline{x}(t) = \epsilon^{(t)}(\frac{\overline{x}(t)}{\sqrt{\sigma^2+1}})d\sigma(t). \tag{3}$$

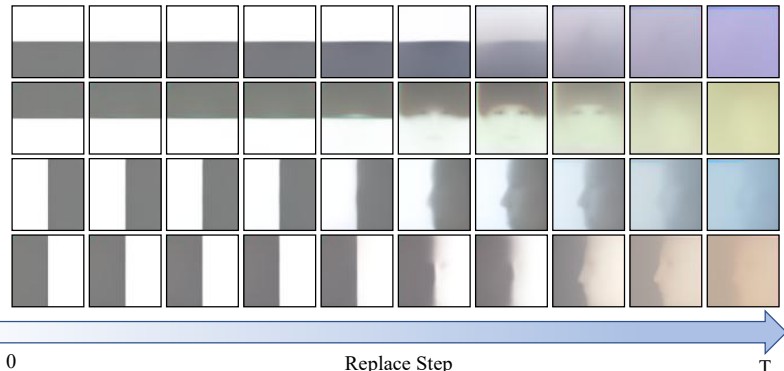

Figure 2: Illustration of *"diffusion in diffusion"*. We inverse pictures of pure gray and white back to $\mathbf{x_t}$, merge them together with different layouts, and then regenerate them back to $x_0$ via deterministic denoising. Different columns indicate different replacement steps $t$. The resulting images show how regions within an image diffuse and interfere with each other during denoising.

By solving this ODE using Euler integration, we can inverse the real image $x_0$ (the visual condition) to $\mathbf{x_t}$ in any corresponding latent space $\epsilon_t$ (Zhu et al., 2023a; Mokady et al., 2023; Asperti et al., 2023) while preserving its information. The symbols $\sigma$ and $\overline{x}$ are the reparameterizations of $(\sqrt{1-\alpha}/\sqrt{\alpha})$ and $(\mathbf{x}/\sqrt{\alpha})$ respectively.

## 3.2 Diffusion in Diffusion

**Internal Interference in Diffusion Generation.** Diffusion in physics is a phenomenon caused by random movements and collisions between particles. The diffusion model, drawing inspiration from non-equilibrium thermodynamics, establishes a Markov chain between a target data distribution and the Gaussian distribution. Subsequently, it learns to reverse this diffusion process, thereby constructing the desired data samples from the Gaussian noise. This inherently simulates a gradual, evolving process that can be viewed as a random walk through a large number of possible data distributions, which eventually gradually approaches the real data distribution. Therefore, the diffusion models share a similar interference phenomenon as the physical diffusion, characterized by continuous information exchange within the data, ultimately achieving harmonious generation results.

We design a toy experiment to reveal this phenomenon more intuitively. To start with, we apply DDIM Inversion to convert gray and white images into various latent codes along the diffusion timeline, spanning from the start ($t = T$) to the final state ($t = 0$). Existing literatures (Song et al., 2020a; Zhu et al., 2023a) demonstrate that those intermediate latent codes can well reconstruct the raw image via deterministic denoising. In other words, both latent codes contain information inherited from their respective raw images, i.e., pure gray and white colors. Consequently, at each selected time step $t$, we merge half of latent codes from the two images into one and denoise the resulting combination. This allowed us to observe how different pieces of information interact throughout the generation process and thus influence the final image. The results in Fig. 2 show that as the merging time step $t$ (at which step we put two latent codes together) approaches $T$ (the Gaussian noise end), the corresponding denoised image $x_0$ exhibits the spatial diffusion phenomenon, resulting in stronger color blending. Conversely, as $t$ approaches $0$ (the raw image end), the image showcases robust reconstruction ability, with minimal interference between the two colors.

**Varying Interference Intensity in Diffusion Generation.** Based on the observations above and additional supplementary experiments in Appendix A.4, we can roughly divide the denoising process into three stages. Throughout the reverse diffusion process, the model attends to different levels of information at each stage, essentially embodying a progression from extreme noise to semantic formation to refinement. Introducing guidance too early hinders its reflection in the final image due to the uncontrollable interference caused by excessive noise, while introducing it too late does not allow for the desired high-level semantic modifications. The best to inject visual condition information is the middle stage, where the model gains the capacity to comprehend and generate basic semantic content, striking a balance between controllable inner mutual influence and responsiveness to text conditions. Ultimately, proper refinement at the last stage ensures that the generated

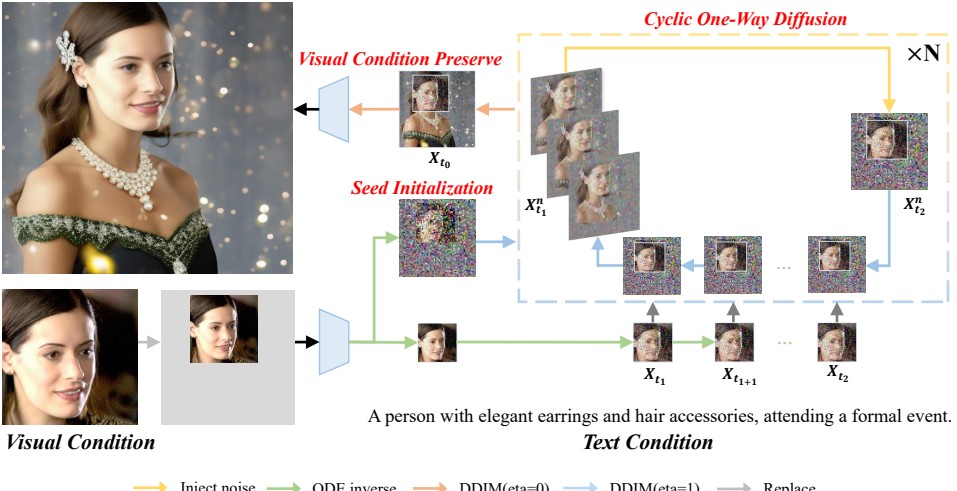

Figure 3: The pipeline of our proposed COW method. Given the input visual condition, we stick it on a predefined background and inverse it as the seed initialization of the starting point of the cycle. In the Cyclic One-Way Diffusion process, we "disturb" and "reconstruct" the image in a cyclic way and ensure a continuous one-way diffusion by consistently replacing it with corresponding $x_t$.

images exhibit intricate details of visual condition. These insights and observations pave the way for integrating new visual condition control paradigms into pre-trained diffusion models.

## 3.3 TRAINING-FREE CYCLIC ONE-WAY DIFFUSION

In this subsection, we illustrate how to take advantage of the diffusion phenomenon in the diffusion generation process, to enable effective both pixel-level and semantic-level visual conditioning without training. Our approach involves three main components: Seed Initialization, Cyclic One-Way Diffusion and Visual Condition Preservation, as shown in Fig. 3.

**Seed Initialization.** It is common to bring the thread end close to the needle before threading. Using random initialization sampled from the Gaussian distribution may introduce features or structures that conflict with the given visual condition. For instance, if the user specifies that the object should be located on the left side, but the initial noise tends to generate it on the right side, there will be a conflict. The model must expend considerable effort during the generation process to correct this inconsistency. To address this issue, we propose to introduce user-specified visual conditions at the initial stage by placing them onto a predefined background, typically a semantically neutral pure gray background. The objective is to inject stable high-level semantic information early in the denoising process, effectively reducing the layout conflicts with the visual condition.

**Cyclic One-Way Diffusion (COW).** Theoretically, we invert the visual condition into its latent representation by solving the probabilistic flow ODE (Eq. 3) and embedding it in the initial random Gaussian noise that serves as the starting point, which can provide a good generative prior to maintain consistency with the visual condition. However, the implanted information will be continuously disrupted by inner diffusion from the surrounding Gaussian region at every denoising step. Therefore, we introduced "one-way" and "cyclic" strategies to maximize the flow of information from the visual condition to the whole image and minimize undesired interference from other image regions. To be specific, we store inverted latents of the visual condition at each inversion step in the middle stage (the semantic formation stage), denoted as $x_{t_1}, x_{t_1+1}, \ldots, x_{t_2}$ and gradually embed them in the corresponding timesteps during the generation process. Through this step-wise information injection, we can ensure the unidirectional propagation of information, i.e., it only propagates from the visual condition to the other regions without interference from information in the background or other parts of the image. Given the limited generative capacity of the model at each step, noise is injected to regress the generative process to earlier stages, as illustrated in Equation 1. This cyclic utilization of the model's generative capacity enables the continuous perturbation of inconsistent semantic information, facilitating the re-diffusion of conditional guidance in subsequent rounds. The Cyclic One-Way Diffusion approach benefits the model from a one-way guidance from the

visual condition, creates additional space by cycles for semantic "disturb" and "reconstruct", and ultimately achieves harmony among the background, visual condition, and text condition.

**Visual Condition Preservation.** Conflicts between the visual and the text conditions often exist (such as a smiling face condition and a "sad" text prompt), necessitating a method that can effectively balance these conditions. We observe that the middle stage is still subject to some extent of uncertainty, which in turn leaves enough space for controlling the text condition guidance on the generation of the visual condition region. Meanwhile, in the later stage, the model focuses on refining high-frequency details and textures to enhance image quality while maintaining global structure integrity. Thus we explicitly control the degree of visual condition preservation by replacing the corresponding region at an adjustable step $x_{t_0}$ in the early phase of this later stage. This approach can effectively preserve fidelity to both the visual and the text conditions, achieving harmonious results of style transfer and attribute editing without additional training.

# 4 EXPERIMENTS

We show the versatility and superiority of COW by comparisons with four SOTA baselines under three different task settings. Also, we conduct an exhaustive ablation study to demonstrate the effectiveness of COW utilizing the diffusion phenomenon in machine learning diffusion generation.

## 4.1 EXPERIMENTS SETUP

**Benchmark.** To simulate the visual condition processing in real scenarios, we adopt face image masks from CelebAMask-HQ (Lee et al., 2020) as our visual condition. For the text conditions, we design three kinds of settings: normal prompt, style transfer, and attribute editing. Then we combine them as the dual conditions for our TV2I task.

**Implementation Details.** We implement COW, SD inpainting, and ControlNet on pre-trained T2I Stable Diffusion model (Rombach et al., 2022) sd-v2-1-base, with default configuration condition scale set 7.5, noise level $\eta$ set 1, image size set 512, 50 steps generation (10 steps for COW), and negative prompt set to "a bad quality and low-resolution image, extra fingers, deformed hands". Note that we implement DB and TI following their official code and use the highest supported Stable Diffusion version sd-v1-5. During generation, we set the visual condition size to 256 and randomly chose a place above half of the image to add the visual condition. We choose $x_{t_1}$ to step 25, $x_{t_2}$ to 35, cycle number to 10. We use slightly different settings for the three different tasks. We set $x_{t_3}$ to be [4, 3], eta to be 0 in the normal prompts, $x_{t_3}$ to be 4, eta to be 0.1 in the attribute editing prompts, and $x_{t_3}$ to be 4, eta to be 1 in the style prompts. We perform all comparisons with baselines on 200 images and 300 texts (100 texts for every setting, 1 text to 2 faces in order). We use a single NVIDIA 4090 GPU to run experiments since our proposed COW method is training-free.

**Comparisons.** We perform a comparison with four SOTA works incorporating different levels of the visual condition into pre-trained T2I models: DreamBooth (Ruiz et al., 2022) based on the code[1], TI (Gal et al., 2022), SD inpainting (Rombach et al., 2022), and Controlnet on the canny edge condition (Zhang & Agrawala, 2023). *DreamBooth* (Ruiz et al., 2022) introduces a rare-token identifier along with a class-prior for a more specific few-shot visual concept by finetuning pre-trained T2I model to obtain the ability to generate specific objects in results. *TI* (Gal et al., 2022) proposes to convert visual concept to word embedding space by training a new word embedding using a given visual condition, and uses it directly in the generation of results of specific objects. *ControlNet* (Zhang & Agrawala, 2023) incorporate additional visual conditions (e.g., canny edges) by finetuning a pre-trained Stable Diffusion model in a relatively small dataset (less than 50k) in an end-to-end way. *SD inpainting* (Rombach et al., 2022) preserves the exact pixel of the visual condition when generating an image, and it is the inpainting mode of the pre-trained Stable Diffusion.

**Evaluations.** To evaluate the quality of the generated results for this new task, we consider two aspects of assessment: visual fidelity and text fidelity. Following the same evaluation metrics as in previous works (Gal et al., 2022; Ruiz et al., 2022; Dong et al., 2022), we utilize prestrained CLIP (ViT-B/32) to evaluate the average pairwise cosine similarity between CLIP embeddings of generated images and the prompts, denoted as *CLIP-T*. Additionally, we adopt a face detection

---

[1]https://github.com/ShivamShrirao/diffusers/tree/main/examples/dreambooth

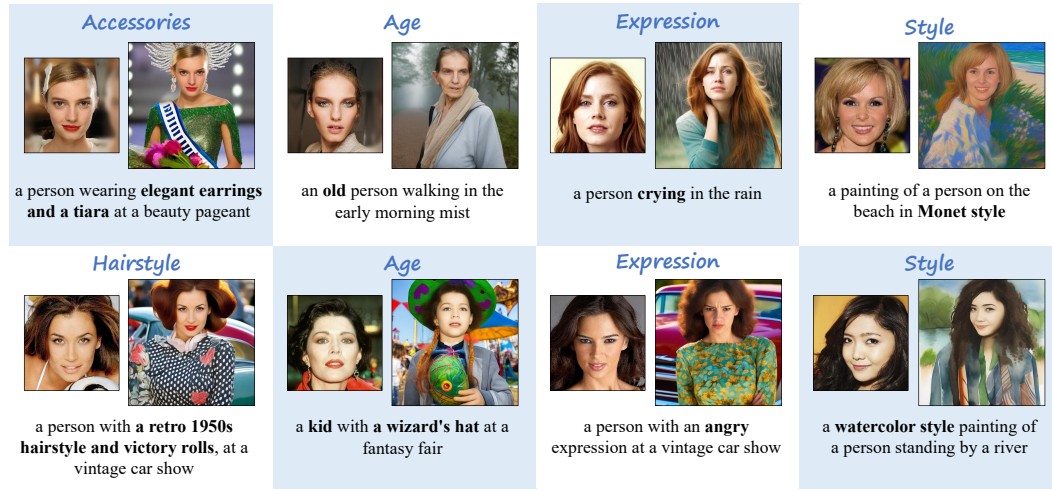

Figure 4: The adaptation of the visual condition to align with the text condition while maintaining the semantic and pixel-level information of the visual condition. In each pair of images, the smaller image is the given visual condition and the other is the generated result. The bolded parts of the text conditions highlight the conflicts between conditions.

Table 1: Quantitative and qualitative comparison between COW and SOTA methods.

| Methodology | Clip-T ↑ | ID-Distance ↓ | Face Detection Rate ↑ | Time ↓ | Condition Consistency ↑ | General Fidelity ↑ |
|---|---|---|---|---|---|---|
| TI | 0.253 | 1.186 | 70.66% | 3025s | 02.73% | 12.60% |
| DreamBooth | 0.329 | 1.361 | 70.50% | 732s | 09.60% | 28.73% |
| ControlNet | 0.305 | 1.194 | 45.66% | 4s | 11.60% | 04.73% |
| SD inpainting | 0.300 | 0.408 | 100.00% | 5s | 06.33% | 02.07% |
| COW (ours) | 0.306 | 0.901 | 100.00% | 6s | **69.73%** | **51.87%** |

model (MTCNN (Zhang et al., 2016)) to detect the presence of the face, and a face recognition model (FaceNet (Schroff et al., 2015)) to obtain the face feature and thus calculate the face feature distance between the generated face region and the given visual condition. The *Face Detection Rate* and *ID-Distance* reflect the fidelity of the visual face condition for those generative models. However, relying solely on model predictions can not fully capture the subtle differences between images and can not reflect the overall quality of the images (e.g., realism, richness), which are critical crucial for human perception. Therefore, we further evaluate our model via **human evaluation**. We design two base criteria and invite 50 participants to be involved in this human evaluation. The two criteria are: 1. *Condition Consistency*: whether the generated image well matches the visual and textual conditions; 2. *General fidelity*: whether the chosen image looks more like a real image in terms of image richness, face naturalness, and overall image fidelity. It's important to note that when assessing the latter criterion, participants are not provided with textual and visual conditions to prevent additional information from potentially interfering with the assessment process.

## 4.2 EXPERIMENTAL RESULTS

*COW* enables versatile applications under the setting of TV2I tasks, including inpainting, attribute editing, and style transfer. More comparison of generated images are included in Appendix A.1.5.

**Quantitative Results.** Quantitative results in Tab. 1 show that our method almost perfectly retains faces (the given visual condition) in the synthesized images, and achieves second-best text conditional fidelity in an efficient time cost. These qualitative results prove that our method generally outperforms previous works in terms of fidelity to visual and text conditions. When processing semantically complex visual conditions, such as faces, by explicitly considering and incorporating low-level visual details, our approach can motivate the model to generate results that are highly consistent with the original conditions. In addition, our method is *training-free*, thus we can generate the prediction using fewer computations.

**Human Evaluations.** We conduct a preference test where participants selecte their favorite image among a shuffled set, including ours and four compared methods. Each image is evaluated by both

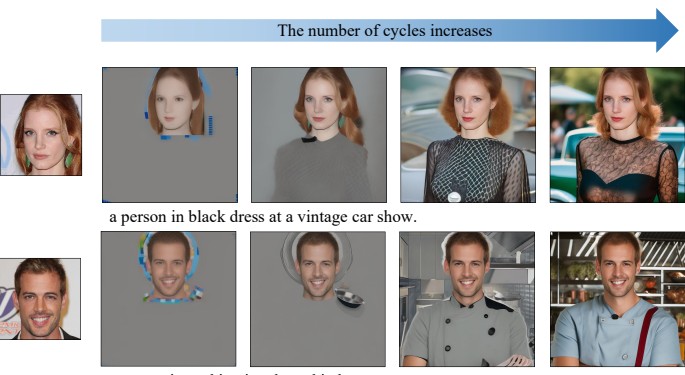

Figure 5: Analysis of the cycling process that diffuses "visual seed" to its surroundings. The left-most figure shows a given face condition. The right shows the images generated with given text conditions. The cycle number increases from the left to the right.

criteria and five different participants, resulting in a total of 3,000 responses for 600 image groups in Tab. 1. Our method consistently outperforms the others across all three settings, as detailed in Appendix A.3. These results demonstrate that our method better integrates text and visual conditions while preserving a satisfying image fidelity.

**One-Way Diffusion during Cycling Process.** To emphasize the role of cyclic one-way diffusion strategy during the generation process, we set the noise level $\eta$ to 0 to slow down the information diffusion. As shown in Fig. 5, as the cycle proceeds, the background pixels are gradually matched to the text condition and fused with the given face condition. This vividly demonstrates that the information in the visual condition keeps spreading and diffusing to the surrounding region along with cycles. In addition, the model is capable of understanding the semantics of the implanted visual conditions properly. Additionally, we conduct a comprehensive ablation study in Appendix A.2 to validate the effectiveness of COW and explore optimal hyperparameter configurations.

**Trade-offs between Conditions for Different Modalities.** The TV2I generation task faces many challenges due to the semantic gap between text and images and the complexity of multimodal data. In general, textual information provides a high-level description of the image to be generated, such as the type, color, and other attributes of the object, while visual information contains more low-level details, such as shape, texture, and so on. The model needs to be able to understand the semantic information in the textual description and translate this information into visually specific details. COW strikes a balance between meeting the visual and text conditions as shown in Fig. 4. For example, when given a photo of a young woman but the text is "an *old* person", our method can make the woman older to meet the text description by adding wrinkles, changing skin elasticity and hair color, etc. while maintaining the facial expression and the identity of the given woman. Additionally, in Fig. 6, we showcase a series of samples containing varying degrees of changes of visual conditions in the generated output: (1) almost unchanged, (2) slightly perturbed (e.g., adding accessories), (3) attribute editing (e.g., from smiling to angry), (4) style transfer (e.g., from a photo to a comic picture), and (5) cross-domain transformation (e.g., from a human face to a lion). In particular, even in cases involving significant conflicts, such as transitioning between different species, our method can adeptly preserve certain individual characteristics of the given individual while seamlessly integrating them with the attributes of the target species as shown in Fig.8. These results demonstrate the ability of COW to effectively understand and balance the information of different modalities and adaptively adjust to produce high-quality images under a wide range of conditions, showcasing its versatility and effectiveness in handling diverse customization scenarios.

**More Applications.** We directly apply COW to other applications to demonstrate its generalization ability. As shown in Fig. 7 (a), we present results involving a generalized visual condition (cats) paired with different text conditions. Our method harmoniously grows a whole body out of the visual condition under the guidance of the text condition. Furthermore, we explore extreme cases where the visual condition size equals the whole image size as illustrated in Fig. 7 (b). The results indicate that our method can still produce pleasant outcomes and maintain its style transfer and attribute editing capabilities even in the context of whole-image generation. Additionally, we include results under multiple visual conditions (*e.g.*, two faces) in Appendix A.1.2.

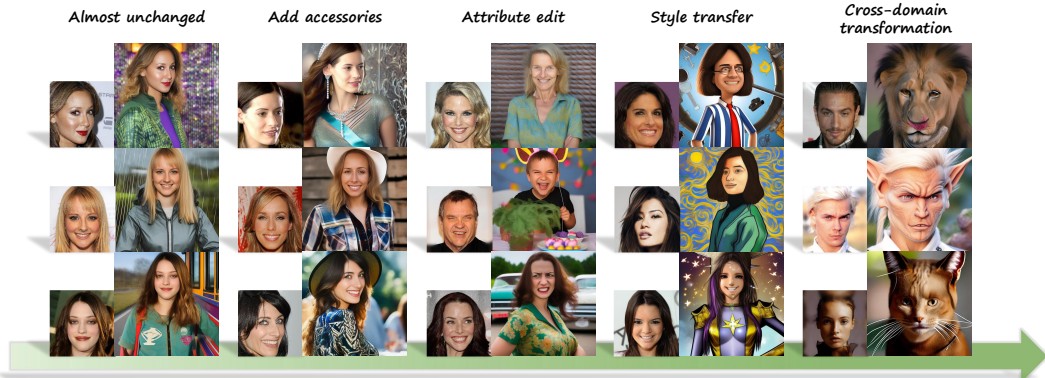

Figure 6: Different degrees of changes in visual conditions. The small image on the left shows the given visual condition and the corresponding generated result is on the right.

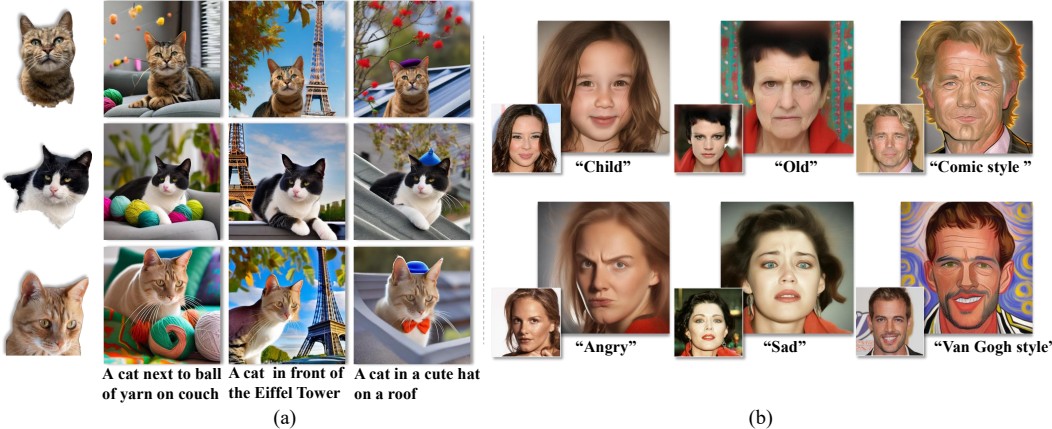

Figure 7: More applications: (a) generalized condition and (b) whole image generation/editing. The small images represent the visual conditions, while the texts serve as prompts.

## 5 CONCLUSION AND DISCUSSION

In this paper, we investigate the diffusion (physics) in diffusion (machine learning) properties and propose our *Cyclic One-Way Diffusion* (COW) method to strict the bidirectional diffusion into a unidirectional diffusion from the given visual condition via a pre-trained frozen diffusion model, fertilizing a versatility of customization application scenarios. Our method novelly explores the potential of utilizing the intrinsic diffusion property for specific task needs. All the experiments and evaluations demonstrate our method can generate images with high fidelity to both semantic-text and pixel-visual conditions in a training-free, efficient, and effective manner.

**Limitations.** The pre-trained diffusion model is sometimes not robust enough to handle extremely strong conflicts between the visual and the text conditions. For example, when given the text "a profile of a person", and a front face condition, it is very hard to generate a harmonious result that fits both conditions. In this case, the model would follow the guidance of the text generally.

**Social Impact.** Image generation and manipulation have been greatly used in art, entertainment, aesthetics, and other common use cases in people's daily lives. However, it can be abused in telling lies for harassment, distortion, and other malicious behavior. Too much abuse of generated images will decrease the credibility of the image. Our work doesn't surpass the capabilities of professional image editors, which adds to the ease of use of this proposed process. Since our model fully builds on the pre-trained T2I model, all the fake detection works distinguishing the authenticity of the image should be able to be directly applied to our results.

ACKNOWLEDGEMENT

This work was partially supported by the National Natural Science Foundation of China under Grant 62372341. This work is jointly advised by Dr. Ye Zhu and Dr. Yu Wu. Also, Ye and Yu appreciate the support from their postdoc advisor, Dr. Olga Russakovsky from Princeton University, for her help in their early career development stage.

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

## A APPENDIX

In the supplementary material, Appendix A.1 showcases an array of TV2I generation results along with comprehensive analyses. We delve into the flexible adaptation of visual conditions under text guidance (Appendix A.1.1), presenting additional TV2I outcomes involving scenarios with multiple faces (Appendix A.1.2). We explore the effects of varying sizes of visual conditions (Appendix A.1.3) and examine failure cases (Appendix A.1.4). Additionally, we conduct extensive image comparisons with baseline methods (Appendix A.1.5). Our thorough ablation study in Appendix A.2 validates the effectiveness of Seed Initialization and provides insights into optimal hyperparameter configurations. Detailed human evaluation results are elaborated upon in Appendix A.3. Moreover, we illuminate the varying sensitivity to text conditions during the denoising process (Appendix A.4.1) and elucidate the semantic formation process of diffusion generation (Appendix A.4.2). Concluding the supplemental material, we furnish the pseudo-code of our proposed pipeline in Appendix A.5.

### A.1 TV2I GENERATION RESULTS COMPARISON AND ANALYSIS

#### A.1.1 DEGREE OF CHANGES ON VISUAL CONDITIONS

COW offers a flexible change in the visual condition region according to the text guidance. The level of changes that may occur within the seed image depends on the discrepancy between textual and visual conditions as shown in Fig. 6. We show a set of samples containing gradual changes of the generated sample: (1) almost unchanged, (2) slightly perturbed (e.g., add accessories), (3) attribute variation (e.g., from smile to angry), (4) style transfer (e.g., from photo to comic picture), and (5) cross-domain transformation (e.g., from human face to lion). The results demonstrate that the seed images can be well preserved given no explicit conflict between the visual-text conditioning pairs, while appropriate levels of text-guided changes can also be well reflected in the case of attribute variation, style transfer, and cross-domain transformation.

We further provide examples with stronger conflicts (cross-domain) in Fig. 8 to demonstrate the model could even fit in the case where the visual condition has a very strong conflict with other conditions. The samples clearly show our method could fit in different conflict levels of text and visual conditions and can handle strong conflicts between conditions.

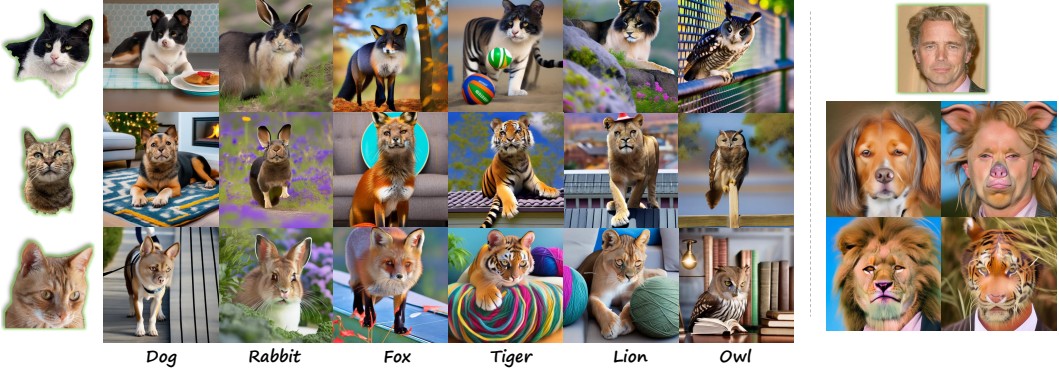

Figure 8: Generated results of cross-domain transformation. Images with green edges are visual conditions.

#### A.1.2 MULTIPLE VISUAL CONDITION RESULTS

We notice that the customized generation methods like DreamBooth (Ruiz et al., 2022) or TI (Gal et al., 2022) learn each object concept separately, which makes the direct generation of multi-target visual conditions infeasible. Additionally, other approaches (Kumari et al., 2022; Avrahami et al., 2023; Gu et al., 2024), that focus on multi-concept customization are often limited to combining 2-3 semantically distinct concepts, and when generating subjects with high similarities, they tend to become confused. Therefore, we try a more difficult setting of two human face conditions in one

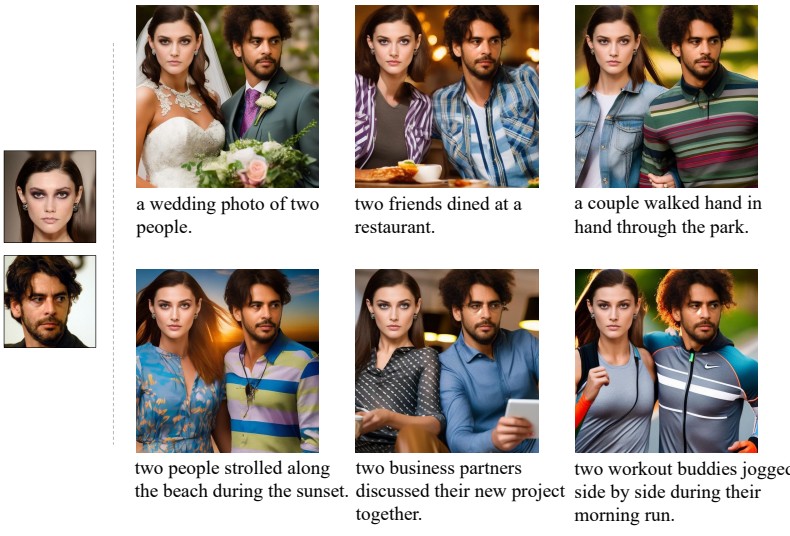

Figure 9: Generated results using two faces as the visual conditions.

diffusion generation and provide a set of results in Fig. 9. As it can be seen from the figure, our method could be easily extended to the multiple visual condition setting.

### A.1.3 DIFFERENT SIZES OF VISUAL CONDITION

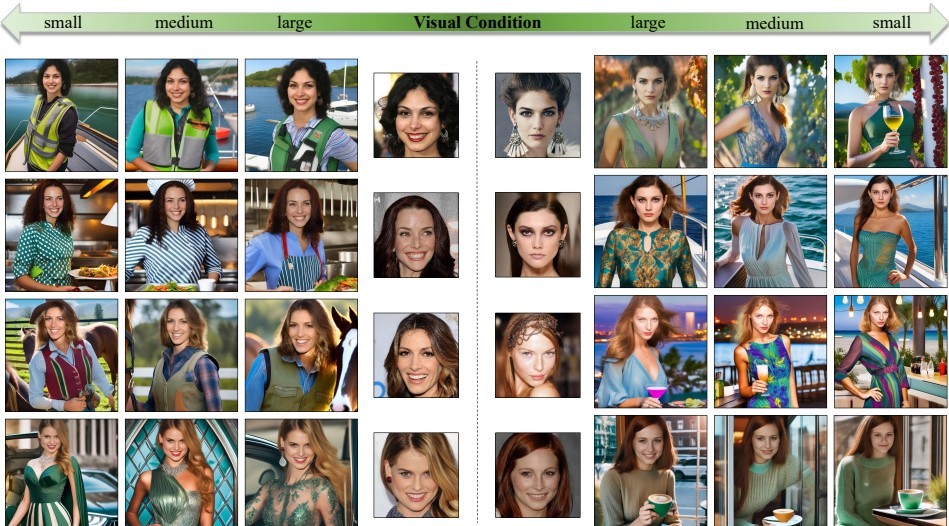

Figure 10: Generated results from different sizes of visual condition. We experimented with Seed Initialization for different sizes of large, medium, and small, respectively.

We directly change the face size of the visual condition applied in the original process. As shown in Fig. 10, although the smaller visual condition is more susceptible to the background, our method still successfully delivers visual condition information and maintains satisfactory results.

### A.1.4 ANALYSIS OF FAILURE CASES

We find the proposed method may fail in two cases as shown in Fig. 11. First, there is a strong orientation mismatch between the visual condition face and the text prompt (e.g., back view text prompt but with front view face visual condition). Second, the model may fail when there is a clear

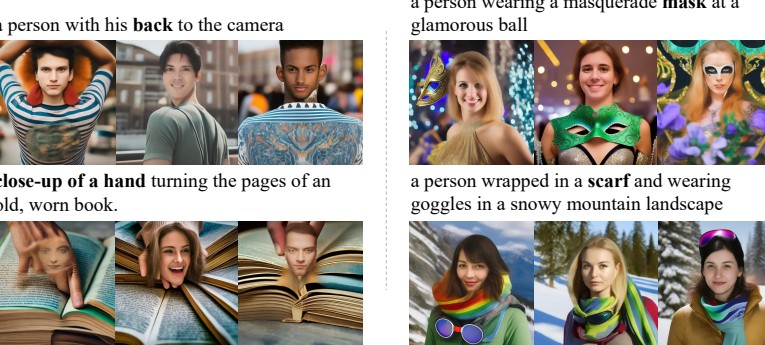

Figure 11: Some failure cases of COW.

text instruction of face coverings while the visual condition is an unobstructed face. In this case, the model may generate the covering at other random positions in the image.

### A.1.5  MORE IMAGE COMPARISONS WITH BASELINES

In Fig. 12, we include a more intuitive comparison with baseline results. We compare current SOTA work that incorporates visual conditions into pre-trained T2I models: DreamBooth (Ruiz et al., 2022), TI (Gal et al., 2022), ControlNet (Zhang & Agrawala, 2023), and SD inpaint (Rombach et al., 2022)

### A.2  MORE ABLATION STUDIES

We ablate on different settings of Seed Initialization (SI) to explore the best setting for SI and verify the effectiveness of SI, as shown in Fig. 13.

**Different Initialization Position.** We show the ablation analysis on Seed Initialization (SI) to see its effectiveness in Fig. 13. The first three columns show different initialization positions for SI, which simulate user customization of visual conditions. We observe that our model can freely generate reasonable images given the pre-defined initialization position.

**Different Seed Initialization Background.** We also try white, black, and random noise backgrounds as the pre-defined background images for SI. The results show using these backgrounds could lightly decrease the quality of generated images compared to the ones in the gray (default) background. We argue this originates from the fact that the average color value (gray) is a better starting point for all possible synthesized images.

**Ablation of Seed Initialization.** The results w/o Seed Initialization exhibit an unnatural merging between background and visual condition, as another person emerges separately from the visual condition. This clearly shows the effectiveness of our SI. Without SI, we can not pre-generate the image layout during the Chaos Phase, where the model only fits the textual condition without considering the visual condition. For example, the model may already generate a person at other positions rather than the user's pre-defined location, which is nearly impossible for us to meet the strict visual condition restrictions given such a layout.

**Ablation of different hyper-parameters.** We conduct a quantitative ablation study on different hyper-parameters of COW, including the number of cycles, or positions of the start and end points in Tab. 2. Here we use three quantitative metrics to analyze different models. *CLIP-T* is the CLIP-text cosine-similarity, which is the similarity of generated image and text condition calculated by the prestrained CLIP (ViT-B/32) model. We also perform human evaluations on those ablation models with 10 volunteers to better evaluate the visual quality of the generated samples. (Each volunteer carefully goes through 216 images generated by the six model variants and rates the best model variant as 5 and the worst as 0, similar to the Mean of Score test in previous works Zhu et al. (2022). Here we report the averaged ratings for each model.) We also include the time cost per image generation. Since different starting and ending points would involve different inversion steps, the time cost is slightly different according to the diffusion inversion cost. The results show the model

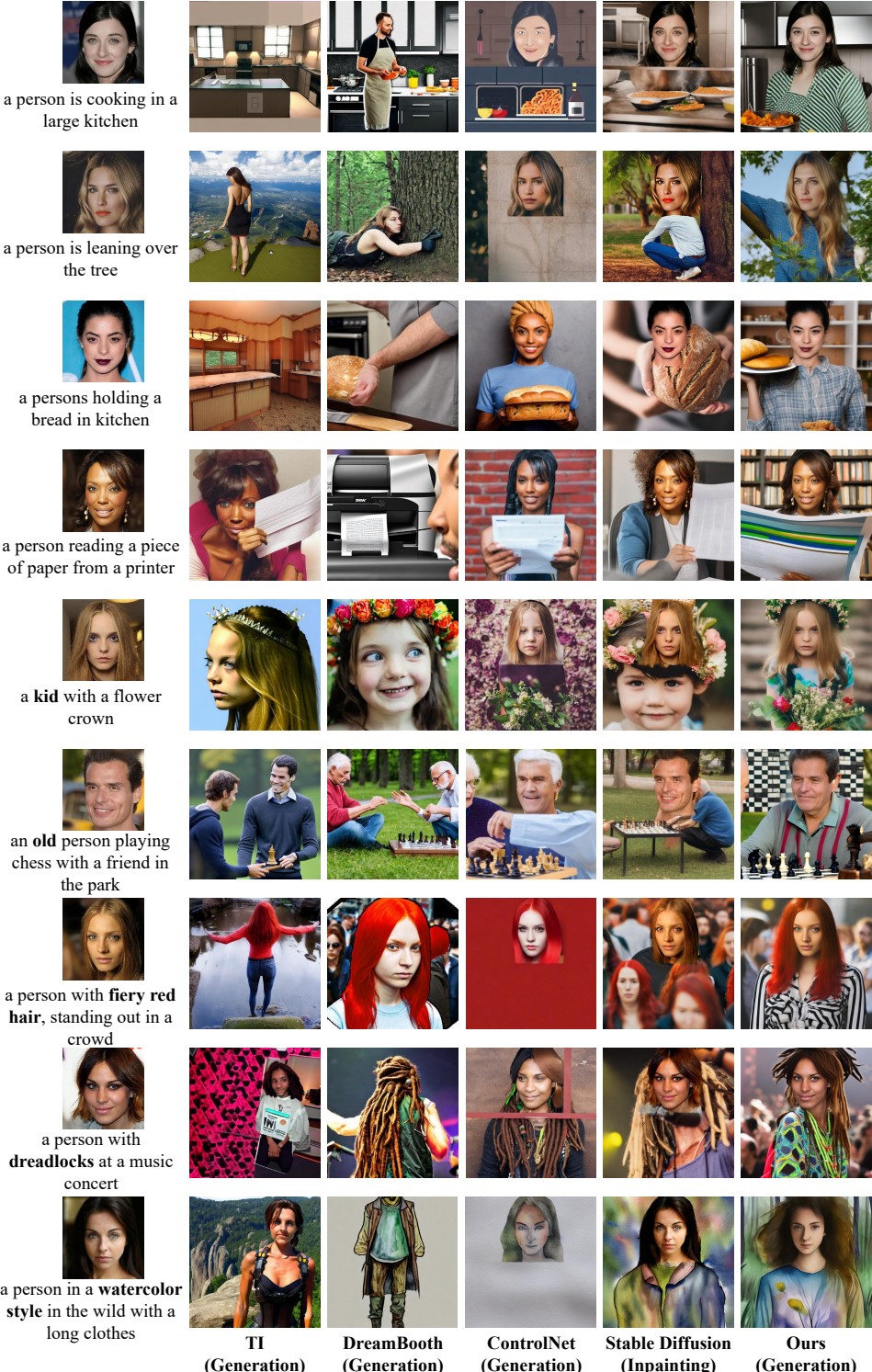

Figure 12: Comparison of our COW-generated images with TV2I baseline.

performance is sensitive to the starting and ending points, which confirms our motivation that we should repeatedly go through the semantic formation stage to have a controllable and directional generation. As we can see, a cycle number of 10 is sufficient and achieves a good balance between performance and speed for the inner diffusion process.

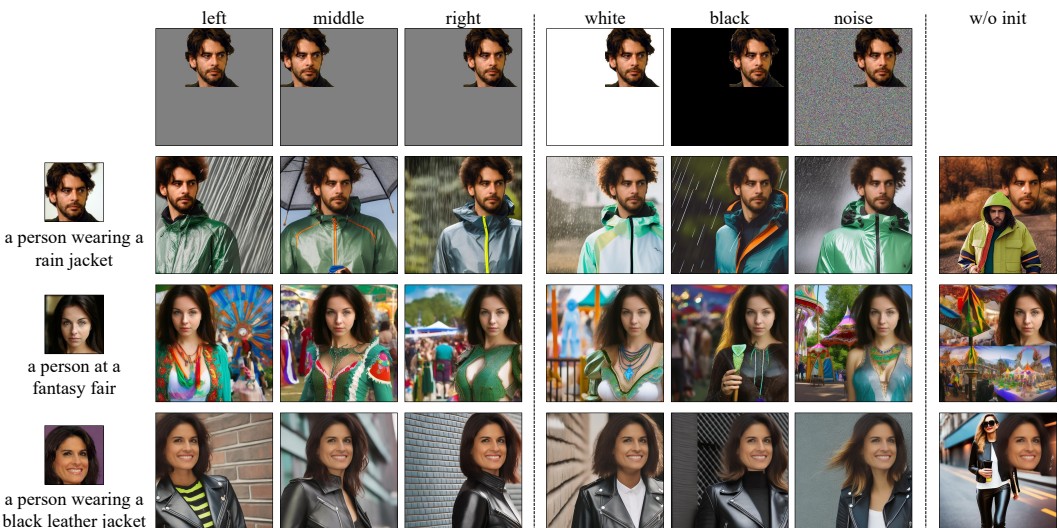

Figure 13: Ablation study on Seed Initialization. Given the left-most visual and text conditions, we show generated images at different initialization positions (left, middle, and right), with different initial backgrounds (gray, white, black, and random noise), and without Seed Initialization.

| Cycle | Position | Clip-T↑ | Human Rating↑ | Time Cost↓ |
|-------|----------|---------|---------------|------------|
| 10 | 700⇒500 | 0.31 | 4.6 | 5.21 |
| 10 | 900⇒700 | 0.30 | 1.5 | 5.52 |
| 10 | 800⇒400 | 0.30 | 3.2 | 7.96 |
| 10 | 500⇒300 | 0.27 | 0.8 | 4.85 |
| 5 | 700⇒500 | 0.29 | 2.5 | 3.32 |
| 20 | 700⇒500 | 0.30 | 3.6 | 8.83 |

Table 2: Ablation of different hyper-parameters of the cycle number and the cycle position using three metrics including Clip-T, Human Rating, and Time Cost.

## A.3 HUMAN EVALUATION

We conduct a human evaluation based on two criteria: 1) Condition Consistency: whether the generated image well matches both the visual and textual conditions; 2) General Fidelity: whether the chosen image looks more like a real image in terms of image richness, face naturalness, and overall image fidelity. It is worth noting that we inform the participants of the application setting: input the face text condition pairs and then get an output image, and ask them to choose a favorite result if they were the users under the two basic criteria. The voting rate for our results greatly surpasses the other baselines in both criteria across all three settings, as shown in Tab. 3, and this demonstrates the general quality when applied to TV2I generation. We show interface examples of questionnaire systems received by participants regarding these two criteria in Fig. 14 and Fig. 15.

The participants recruited are all undergraduate or graduate students, all over 18 years old, ranging from 18-26 years old. Therefore, there are no ethical concerns for young participants (under 18 years

Table 3: The condition correspondence and general fidelity results of human study for three settings: normal prompt, style transfer, and attribute editing, with condition correspondence to the left of every column and general fidelity to the right. Note that our method outperforms all of the baselines in terms of the general fidelity shown by the results of human users' judgment.

|  | TI | | DB | | ControlNet | | SD inpainting | | COW | |
|---|---|---|---|---|---|---|---|---|---|---|
| normal | 2.00% | 10.60% | 2.80% | 21.40% | 5.40% | 3.00% | 5.20% | 2.00% | **84.60%** | **63.00%** |
| style | 3.60% | 15.40% | 13.40% | 26.80% | 24.20% | 7.20% | 10.60% | 3.00% | **48.20%** | **47.60%** |
| editing | 2.60% | 11.80% | 12.80% | 37.80% | 5.40% | 4.00% | 3.20% | 1.20% | **76.00%** | **45.20%** |

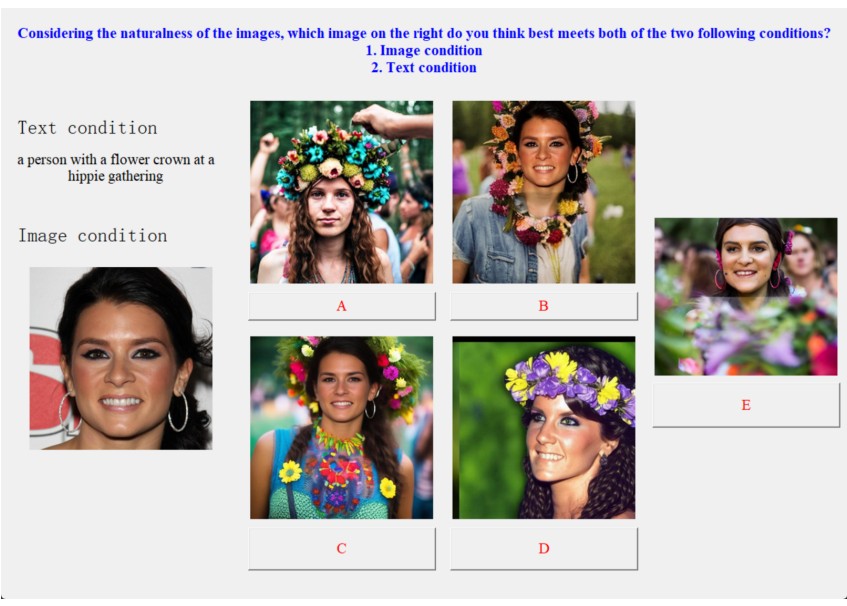

Figure 14: The interface of our human study on conditional consistency.

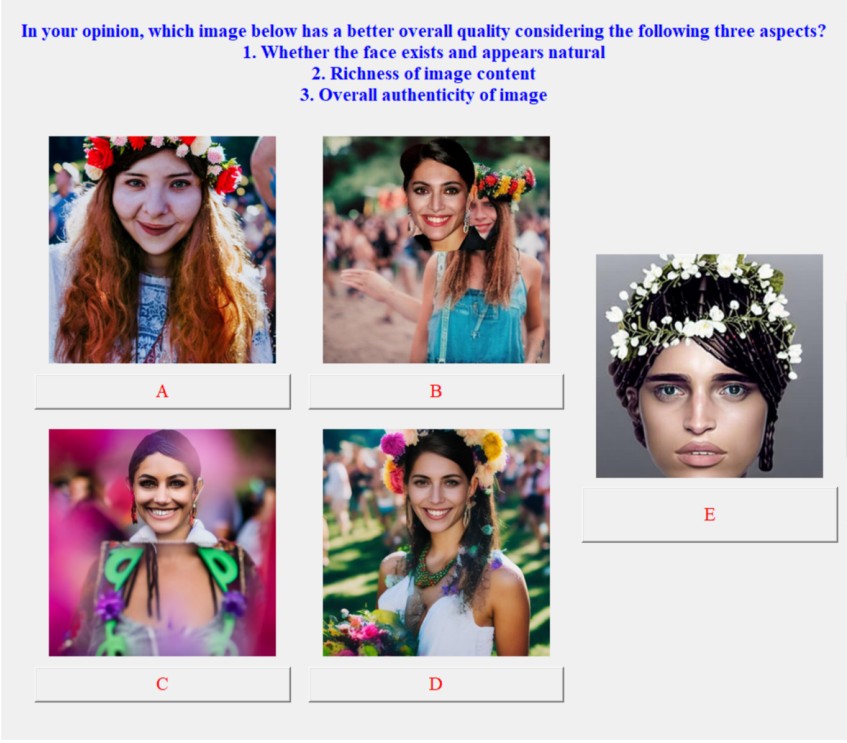

Figure 15: The interface of our human study on general fidelity.

old). Before taking the questionnaire, all participants are informed about the research's purpose, the compensation, and the user study content. All participants consent to participate. We do not provide compensation to volunteers. In addition, all elements of the questionnaire are free from potential psychological or moral hazards.

## A.4 PROPERTIES OF DIFFUSION GENERATION PROCESS

### A.4.1 SENSITIVITY TO TEXT CONDITION DURING DENOISING

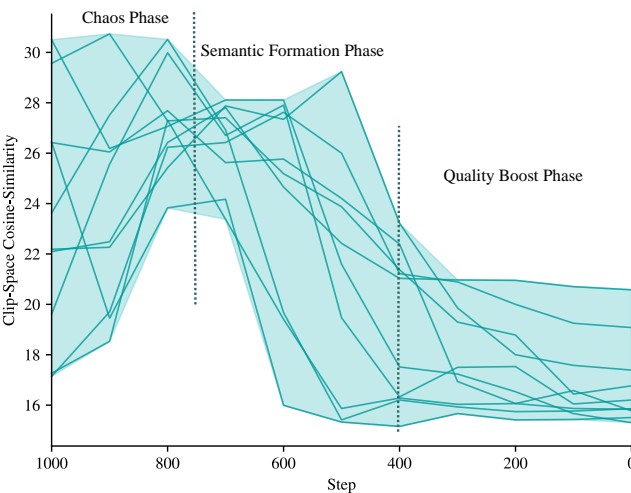

Figure 16: Text-sensitivity during denoising process. Each line represents a CLIP cosine similarity between the generated image and text with the text condition injected at different steps. The image is generated in 1000 overall unconditional denoising process with 100 steps text conditional guidance starting from $t$. Generally, the denoising process is more responsive to the text condition in the beginning and almost stops reacting to the text condition when high-level semantics are settled.

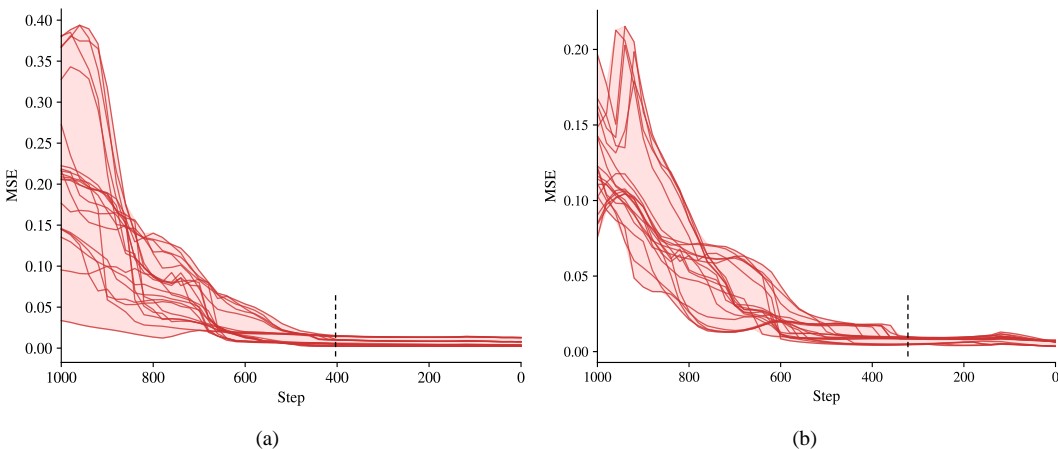

Figure 17: Different sizes come with different semantic formation processes. Each red curve represents a face disturb-and-reconstruct process, (a) size is $256 \times 256$ and (b) size is $128 \times 128$. We disturb the reconstruction process by sticking the origin image to a random noise background ($512 \times 512$) at different steps. The general semantic is settled earlier when the size is larger.

We study the sensitivity of DMs to text conditions during the denoising process by injecting text conditions at different denoising steps. We replace a few unconditional denoising steps with text-conditioned ones during the generation of image random sampled from Gaussian noise. We calculate the Clip cosine similarity between the final generated image and text condition. As shown in Fig. 16, the model is more sensitive to text conditions in the early denoising stage and less sensitive in the late stage. This also demonstrates our method reasonably utilizes the sensitivity to text conditions through proposed Seed Initialization and Cyclic One-Way Diffusion.

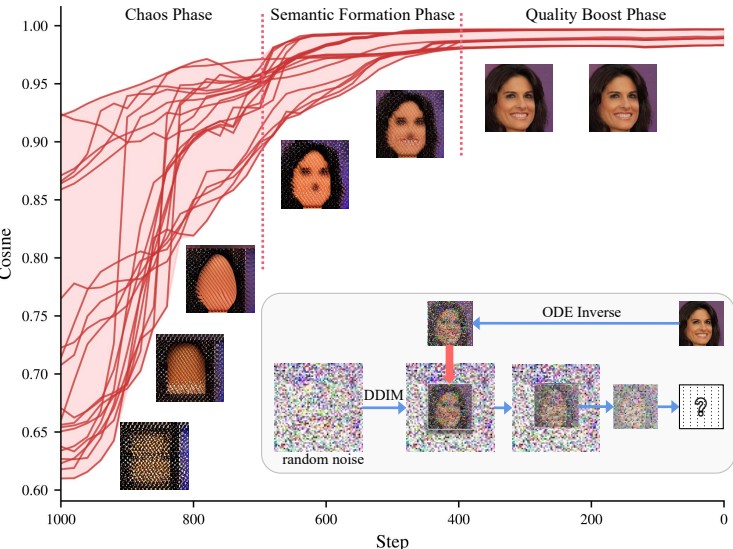

Figure 18: Illustration of the semantic formation process and the mutual interference. Each red curve represents a face disturb-and-reconstruct process. We only show one face for instance. We disturb the reconstruction process by sticking the origin image to a random noise background at different steps. The cosine similarity between the original image and the reconstructed image increases as the denoising step goes.

### A.4.2 ILLUSTRATION OF SEMANTIC FORMATION PROCESS OF DIFFUSION GENERATION

To quantitatively assess the influence between regions during the denoising process, we inverse an image $x_0$ to obtain different transformed $\mathbf{x_t}$, and plant it in a replacement manner into a sub-region of random Gaussian noise in the same step. We subsequently apply this composite noise map to several denoising steps before extracting the planted image to continue the generation process. As shown in Fig. 18, the influence level weakens with the denoising process.

To see the degree of impact exerted on images of different sizes during the denoising process, we conduct the same experiments with images of two different sizes ($128 \times 128$, $256 \times 256$). We inverse images to $\mathbf{x_t}$, and reconstruct them with disturbance introduced by sticking them to a random noise background ($512 \times 512$) at the corresponding step for 100 steps. We calculate the MSE loss of the original image and the final reconstructed image after being disturbed. As shown in Fig. 17, the semantics settle earlier when the size is larger.

A.5  PSEUDO CODE

We show the pseudo-code for our proposed Cyclic One-Way Diffusion approach in Algo. 1.

---

**Algorithm 1:** Cyclic One-Way Diffusion

---

**Input:** visual condition $V_0$, text prompt $P$, target cycle number $c$, ends of the cycle $t_1, t_2$, id preserve step $t_3$
**Output:** Generated Image $x$
**Definition:** deterministic sampling $p_s$, stochastic sampling $p_d$
**// Step 1: Seed Initialization**
$X_0 \leftarrow$ paste $V_0$ into the user-specified background by the user-specified size and position;
$x_0, v_0 \leftarrow \texttt{Encoder}(X_0), \texttt{Encoder}(V_0)$;
$x_{t_1}, v_t \leftarrow \texttt{ODEInverse}(x_0, t_1), \texttt{ODEInverse}(v_0, t)$;
**// Step 2: Cyclic One-Way Diffusion**
**while** $cyc_{num} < c$ **do**
    **for** $t = t_1, t_1 - 1, ..., t_2$ **do**
        $x'_t \leftarrow \texttt{Replace}(x_t, v_t)$;
        $x_{t-1} \leftarrow p_s(x'_t, t, P)$;
    **end**
    $x_{t_1} \leftarrow \texttt{InjectNoise}(x_{t_2})$;
**end**
**// Step 3: Visual Condition Preserve**
**for** $t = t_2, t_2 - 1, ..., 0$ **do**
    **if** $t == t_3$ **then**
        $x'_t \leftarrow \texttt{Replace}(x_t, v_t)$;
        $x_{t-1} \leftarrow p_d(x'_t, t, P)$;
    **end**
    $x_{t-1} \leftarrow p_d(x_t, t, P)$;
**end**

---

