# OpenReview forum: "Diffusion in Diffusion: Cyclic One-Way Diffusion for Text-Vision-Conditioned Generation"
_ICLR.cc/2024/Conference — ICLR 2024 poster_

### Official Review · Reviewer_45wn · 2023-10-31

**Soundness:** 3 good
**Presentation:** 3 good
**Contribution:** 3 good
**Rating:** 5
**Confidence:** 4

**Summary:**

This manuscript proposes a new method called Cyclic One-Way Diffusion that integrates diffusion in physics and diffusion in deep learning, providing a learning-free manner by controlling the direction of diffusion in various customization application scenarios.

**Strengths:**

1. The article proposed a novel method from a new perspective to utilize the capabilities of the diffusion model.
2. A learning-free manner can be widely used in personality customization with one or several conditions.
3. The experiment results show good performance.

**Weaknesses:**

1. The approach lacks a theoretical foundation, so it is not very intuitive to express why it can work.
2. The results of comparison methods are a bit too bad. More information about the setting should be given.

**Questions:**

I have some questions for the author to further improve this work.
1. For the consideration of reproducibility, the code of the proposed method is suggested to be provided.
2. Are the comparison methods learning-free? If so, it would be beneficial to provide additional details about the experimental settings. Furthermore, for the comparison experiments, it might be advisable to incorporate some learning-free methods, such as LIVR [1], to ensure a comprehensive evaluation.
3. Sections 3.2 and 3.3 lack a theoretical foundation or pseudo-code to facilitate clearer understanding and reading.
4. It is not clear in Section 3.3, paragraph 4. What is the difference between two steps/two ends and t-th diffusion step, and how to use them?
5. The evaluation in Table 2 and in Section 4.2 is not matched.

---

> ### Author Response · Authors · 2023-11-19
>
> We sincerely thank you for providing your valuable suggestions and pointing out our oversights! Please see our response below:
> - **Q1:** The approach lacks a theoretical foundation, so it is not very intuitive to express why it can work.
> - **A1 :**
>     - Our method mainly builds upon the observed "diffusion phenomenon" of the pre-trained model as demonstrated in Fig. 2. This phenomenon is inspired by the nature that the diffusion model follows a stochastic Markovian chain between the data distribution and Gaussian distribution, characterized by a gradual entropy increase through Gaussian perturbations. In the reverse diffusion process, models focus on different levels of information at different phases, essentially embodying an extremely-noisy-to-semantic-formation-to-refine process.
>     - The two main proposed manners for utilizing the above properties are "One-Way" control and the "Cyclic" construction.  By constantly replacing in the Semantic Formation Phase, We exert influence from the given visual condition to the rest of the generated image while preserving the content of the visual condition, thus achieving a one-way conditioning. However,  the pretrained model can easily fail to generate a pleasant result with a roughly spliced starting point. So we design a cyclic "disturb and reconstruct" process to recursively exploit the former generation progress under the visual condition, thus achieving better generation results.
>     - Hope this additional explanation can help solve your concerns.
> ---
> - **Q2:** The results of comparison methods are a bit too bad. More information about the setting should be given.
> - **A2:**
>     - We follow the corresponding official codes to conduct all the comparison methods. For ControlNet, we directly use the official webui of Stable Diffusion. We have included codes of other methods in the updated supplementary materials.
>     - The unsatisfying performance of compared customization methods (DreamBooth, TI) on semantically complex conditions like human face has also been reported in prior works [a, b, c]. One major drawback of these methods is that they are not training-free and thus tend to easily overfit the limited finetune data and incorrectly entangle object identity and spatial information. Instead, our work is learning-free, leveraging the property of diffusion denoising itself for the customized generation.
>     - The unsatisfying performance of SD-inpainting and ControlNet comes from the compulsively preserved visual condition, which may easily generate results with significant fragmentation (as shown in Fig. 1 of the main paper). Providing a small area as visual condition is a big challenge for these two methods. Meanwhile, these methods lack flexibility of fitting in multiple conditions, thus cannot be used for attribute editing or style transfer applications where the users use text prompts to slightly revise the visual conditions.
>
> [a] Jing Shi, Wei Xiong, Zhe Lin, and Hyun Joon Jung. Instantbooth: Personalized text-to-image generation without test-time finetuning. arXiv preprint arXiv:2304.03411, 2023.
>
> [b] Yuheng Li, Haotian Liu, Yangming Wen, and Yong Jae Lee. Generate anything anywhere in any scene. arXiv preprint arXiv:2306.17154, 2023.
>
> [c] Ziyi Dong, Pengxu Wei, and Liang Lin. Dreamartist: Towards controllable one-shot text-to-image generation via contrastive prompt-tuning. arXiv preprint arXiv:2211.11337, 2022.

---

> > ### Author Response · Authors · 2023-11-19
> >
> > Here are the responses to the remaining questions:
> >
> > - **Q3:** For the consideration of reproducibility, the code of the proposed method is suggested to be provided.
> > - **A3:** The anonymous code has already been included in the Supplementary Meterial of the initial submission. We will make the code publicly released upon acceptance.
> > ---
> > - **Q4:** Are the comparison methods learning-free? If so, it would be beneficial to provide additional details about the experimental settings. Furthermore, for the comparison experiments, it might be advisable to incorporate some learning-free methods, such as LIVR [1], to ensure a comprehensive evaluation.
> > - **A4:**
> >     - Among the compared four methods, only SD inpainting is learning-free. The other three (DreamBooth, TI, ControlNet) have to be finetuned before generation. For SD inpainting and our model, we input the given visual condition and text prompt at the same time to the model for direct inference. For DreamBooth and TI, we first use the given visual condition to update the model parameters or prompt tokens according to their default training settings, and then generate the target image using the text prompt and finetuned model. ControlNet first trains an extra module on edge images, and then generate the target image using the edge image of the given visual conditons.
> >     - ILVR [a] is essentially a visual conditioned method by utilizing classifier guidance to generate images closer to reference images on face model. However, this work only takes visual conditions as input and generates a random new face that is similar to the given face. Thus this work could not be mitigated to the text-visual customization task (TV2I), where we intend to generate a new image that not only contains the visual condition but also follows the text prompt guidance. The core of this task is to fit in both two visual/textual conditions to generate a harmonious result. As far as we know, we are the first and only learn-free TV2I work so far. We have added these discussions in the related work section of the revised main paper.
> >
> > [a] Jooyoung Choi, Sungwon Kim, Yonghyun Jeong, Youngjune Gwon, and Sungroh Yoon. Ilvr: Conditioning method for denoising diffusion probabilistic models. arXiv preprint arXiv:2108.02938, 2021.
> >
> > ---
> > - **Q5:** Sections 3.2 and 3.3 lack a theoretical foundation or pseudo-code to facilitate clearer understanding and reading.
> > - **A5:**  Thank you for your suggestion. We added a pseudo-code in Sec. A.5 of the revised paper to help clearer understanding.
> > ---
> > - **Q6:** It is not clear in Section 3.3, paragraph 4. What is the difference between two steps/two ends and t-th diffusion step, and how to use them?
> > - **A6:**
> >     - Two steps $x_{t_{1}}$, $x_{t_{2}}$ in paragraph 5 are identical to $x_{t-k}$, $x_{t}$ in paragraph 4. They refer to the starting point (high noise) and ending point (low noise) of our cycle process.  In experiments, we use the starting point t2 of 700-th, and the ending point t1 of 500-th. Since there is a large redundant in the cyclical process, we can apply the skip-step strategy of DDIM to speed up the "reconstruct" process by 100x, i.e., we only use three steps (700-th, 600-th, 500-th) for the "reconstruct" process.
> >     - After the reconstruction part, we further disturb the generation process by changing the noise level back to 700-th (can be viewed as returning back to the starting point of 700-th). A new cycle begins.
> >     - We have updated the corresponding description in the manuscript to make it clear.
> > ---
> > - **Q7:** The evaluation in Table 2 and in Section 4.2 is not matched.
> > - **A7:**  Thanks for pointing out the typo. The 'conditional consistency' in Table 2 and  'conditional correspondence' in Section 4.2 are the same thing.  We have corrected this typo and updated it in the new version.

---

> > > ### Comment · Area_Chair_7EF3 · 2023-12-04
> > > **Dear reviewer**
> > >
> > > Were you able to review the author's rebuttal and the other reviewer's commentary? Do you still hold the same score after reading these?

---

### Official Review · Reviewer_4D12 · 2023-11-01

**Soundness:** 3 good
**Presentation:** 3 good
**Contribution:** 3 good
**Rating:** 6
**Confidence:** 2

**Summary:**

This paper proposed a ``diffusion in diffusion'' approach that leverages both physical diffusion and learning-based diffusion for the text-vision-conditioned generation, e.g., in painting, attribute editing and style transfer. The proposed method is based on inverse seed initialization and "disturb" and "construct" cycles for diffusion.  The proposed method is justified to be able to generate realistic image with higher ID-Distance and Face detection rates.

**Strengths:**

(1) The idea of diffusion in diffusion for diffusing image to generate consistent background content in the diffusion process is an interesting idea, and can be combined with the pre-trained diffusion model without retraining.

(2) The experiments on inpainting with visual condition, text-vision-conditioned generation showed that the proposed approach can produce realistic images.

**Weaknesses:**

The overall idea of this approach is interesting. I have some questions mainly on the experimental justifications as follows.

1. Most of these examples are based on putting an object in bounding box to a large image by adding backgrounds. This setting has applications, however, whether this approach can be applied to whole image generation/editing instead of pasting object on a larger image.

2. In the main body of this paper, the authors should present failure cases if it has, and analyze the reasons.

3. The inner diffusion cycles for "disturb", "reconstruct" may introduce additional computational overhead. More details on the computational balance on the number of cycles and its effect on the results should be given.

4. There are typos in the paper, e.g., t around eq.(1). Please check the whole paper.

5. On page 4, the cycles are divided into three phases. Is there strict division boundary between these phases, and why these discussions are introduced to the main body of this paper, and how to support these conclusions on the existing of three phases.

**Questions:**

Please see above for my questions on the experiments and discussions.

---

> ### Author Response · Authors · 2023-11-19
>
> Thank you for your professional feedback, which played a crucial role in refining our manuscript. Please see our response below:
> - **Q1:** Most of these examples are based on putting an object in bounding box to a large image by adding backgrounds. This setting has applications, however, whether this approach can be applied to whole image generation/editing instead of pasting object on a larger image.
> - **A1:** Thanks for the valuable suggestion! We conduct experiments on the whole image generation/editing, achieving superior results. We have added those samples in the new appendix (Fig. 11 of the appendix).
> ---
> - **Q2:** In the main body of this paper, the authors should present failure cases if it has, and analyze the reasons.
> - **A2:**
>     - Thanks for the valuable suggestion. We have added the failure cases and the analysis in 13 of the revised paper.
>     - We found the proposed method usually failed in two cases. First, there is a strong orientation mismatch between the visual condition face and the text prompt (e.g., back view text prompt but with front view face visual condition). Second, the model may fail when there is a clear text instruction of face coverings but the visual condition is a unobstructed face. In this case, the model may generate the covering at other random positions in the image.
> ---
> - **Q3:** The inner diffusion cycles for "disturb", "reconstruct" may introduce additional computational overhead. More details on the computational balance on the number of cycles and its effect on the results should be given.
> - **A3:**
>     - The inner diffusion cycles for "disturb"and "reconstruct" are only conducted in part of the denoising path (20% of the whole process). The "disturb" step does not include additional computation overhead, since we just change the noise level from 500-th (low noise level) back to 700-th  (high noise level).  In this way, the denoising process can be viewed as returning back to the starting point of 700-th.
>     - Since there is redundancy in the cyclical process, we can apply skip steps strategy of DDIM to speed up the "reconstruct" progress by 100x. As a result, our computation cost is only 20% higher than the original Stable Diffusion inpainting model (6s versus 5s).
>     - To better address this concern, we have conducted more ablation studies to investigate the computation cost and performance of different cycle numbers and starting/ending points. The quantitive results are shown in the table below. As we can see, a cycle number of 10 is sufficient and achieves a good performance/speed balance for the inner diffusion process.
>
> | Cycle | Position &nbsp;&nbsp;&nbsp;           \|  | Clip-T$\uparrow$ | Human Rating$\uparrow$ | Time Cost$\downarrow$ |
> |:-----:|---------------------:|:----------------:|:----------------------:|:---------------------:|
> | 10    | 700$\Rightarrow$500 &nbsp;\| | 0.31             | 4.6                    | 5.21                  |
> | 10    | 900$\Rightarrow$700 &nbsp;\|   | 0.30             | 1.5                    | 5.52                  |
> | 10    | 800$\Rightarrow$400 &nbsp;\|   | 0.30             | 3.2                    | 7.96                  |
> | 10    | 500$\Rightarrow$300 &nbsp;\|   | 0.27             | 0.8                    | 4.85                  |
> | 5      | 700$\Rightarrow$500 &nbsp;\|   | 0.29             | 2.5                    | 3.32                  |
> | 20    | 700$\Rightarrow$500 &nbsp;\|   | 0.30             | 3.6                    | 8.83                  |
>
> ---
> - **Q4:** There are typos in the paper, e.g., t around eq.(1). Please check the whole paper.
> - **A4:** Thanks for pointing it out! We have checked the whole paper and polished the writing again. The updated version has been uploaded in OpenReview.
> ---

---

> > ### Author Response · Authors · 2023-11-19
> >
> > Here is the response to the remaining question:
> > - **Q5:** On page 4, the cycles are divided into three phases. Is there strict division boundary between these phases, and why these discussions are introduced to the main body of this paper, and how to support these conclusions on the existing of three phases.
> > - **A5:**
> >     - There are no strict boundaries, but these phases can be properly traced to a certain range. The absence of a strict boundary is due to the nature of the diffusion model, which inherently simulates a gradual, evolving process (Diffusion model iteratively/gradually transforms data to approach the true underlying data distribution by introducing and removing noise at each step.)
> >     - The transition between distributions is achieved through Gaussian perturbations, reflecting a continuous process, from noisy chaos, to semantic formation, to refinement, which can be approximated by measuring the distance between distributions (more theoretical details can be found in [a] ).
> >     - Introducing guidance during the first phase (Chaos Phase) hinders its reflection in the final image due to the uncontrollable interference caused by excessive noise while introducing it too late during the third phase (Quality Boost Phase) does not allow for the desired high-level semantic modifications. These properties were well demonstrated in Sec. A.3 of Appendix and we can also see clear differences among the three phases in Fig. 15 and Fig. 17.
> >     - We emphasize this part in the main paper, not only because it establishes the foundation of our methodological design, but also conveys a valuable message to the community: the learning-free controlling of generative output can be much more efficient than the current learning-based and step-wise paradigms. Specifically, current methods seek to integrate conditional guidance in a step-wise manner by adding conditional information at every diffusion step throughout the entire denoising process. However, we demonstrate that the above step-wise conditioning is not necessary to achieve fine-grained control of the generative output. Instead, operating on a critical range is sufficient (20% of the entire steps in this case).
> >
> >
> > [a] Ye Zhu, Yu Wu, Zhiwei Deng, Olga Russakovsky, and Yan Yan. Boundary guided mixing trajectory for semantic control with diffusion models. NeurIPS, 2023.

---

### Official Review · Reviewer_S1EF · 2023-11-01

**Soundness:** 3 good
**Presentation:** 3 good
**Contribution:** 3 good
**Rating:** 6
**Confidence:** 4

**Summary:**

This paper proposes a training-free method to better preserve the visual conditions in diffusion-based text-vision-conditioned image generation.

**Strengths:**

1. The paper is well-written and easy to follow. The idea is clearly illustrated.

2. The proposed method is elegant and straightforward to preserve the visual condition.

**Weaknesses:**

1. The proposed method repeatedly replaces part of the diffusion latent variable x_t with the corresponding visual condition, which strongly maintains the visual condition in the generated image. However, this method may have an intrinsic drawback: the visual condition may be too strong and conflict with other conditions. in which case, the generated image may be unrealistic.

2. There are no quantitative analyses of the number of cycles, or positions of the start and end points. These experiments are important for us to understand the effectiveness of the proposed method.

3. According to Table 1, the proposed method is inferior to SD inpainting on both performance and efficiency. The only superiority of the proposed method is training free. However, since it needs cyclical diffusion & denoising, its inference cost is higher than SD inpainting. The superiority may be weakened.

**Questions:**

See the weakness above

------------------------

The authors' response addresses most of the concerns. And I would like to keep my positive rating.

---

> ### Author Response · Authors · 2023-11-19
>
> We greatly appreciate your insightful and detailed feedback. The comments and suggestions made us think more deeply about our method and were very helpful in improving the quality of our work. Please see our point-to-point response below:
> - **Q1:** The proposed method repeatedly replaces part of the diffusion latent variable $x_t$ with the corresponding visual condition, which strongly maintains the visual condition in the generated image. However, this method may have an intrinsic drawback: the visual condition may be too strong and conflict with other conditions. in which case, the generated image may be unrealistic.
> - **A1:**
>     - In fact, the steps of "repeatedly replace" operation of the visual condition only occupy 20% of the generation process, while the text condition works throughout the whole process (from the layout to the detail). Our approach demonstrates a great ability to deal with different levels of conflicts between the visual and the textual conditions in a unified pipeline, as shown in Fig. 1 and Fig. 4 of our original submitted manuscript.
>     - In the meanwhile, we note the reviewer's concern on the visual-textual conflict is more of a problem for general customization tasks, but not specific to the methodological design of COW. Essentially, it is about how we would want a generative model to comprise between controversial given conditions. For instance, existing methods such as DreamBooth may also produce unrealistic images given very different conditioning prompts.
>     - In experiments we confirm COW is still able to handle different levels of conflicts.  In Fig. 7 of the new appendix, we show a gradual change in different conflict levels of textual and visual conditions. COW could even fit in the case where the visual condition has a very strong conflict with other conditions (e.g., cross-domain transformation from human to lion in Fig. 8). This further validates our model has the ability to fit in both conditions at the same time.
> ---
> - **Q2:** There are no quantitative analyses of the number of cycles, or positions of the start and end points. These experiments are important for us to understand the effectiveness of the proposed method.
> - **A2:**
>     - We further conduct an ablation study on different hyper-parameters of COW. The table below provides quantitative analysis on the ablation of different settings of cycle numbers, the position of the starting and the ending points. We also included the results in Table 4 of the revised Appendix.
>     - Here we use three quantitative metrics to analyze different models. CLIP-T is the CLIP-text cosine-similarity, which is the similarity of generated image and text condition calculated by the prestrained CLIP (ViT-B/32) model. To better evaluate the visual quality of the generated samples, we also perform a human evaluation on those ablation models with 10 volunteers. Each volunteer carefully went through 216 images generated by the six model variants, and rated the best model variant as 5 and the worst as 0. Here we report the averaged ratings for each model. Since different starting and ending points would involve different inversion steps, the time cost is slightly difference according to the diffusion inversion cost.
>     - The results show the model performance is sensitive to the starting and ending points, which confirms our motivation that we should repeatedly go through the Semantic Formation phase to have a controllable and directional generation.
>
>     | Cycle | Position &nbsp;&nbsp;&nbsp;           \|  | Clip-T$\uparrow$ | Human Rating$\uparrow$ | Time Cost$\downarrow$ |
>     |:-----:|---------------------:|:----------------:|:----------------------:|:---------------------:|
>     | 10    | 700$\Rightarrow$500 &nbsp;\| | 0.31             | 4.6                    | 5.21                  |
>     | 10    | 900$\Rightarrow$700 &nbsp;\|   | 0.30             | 1.5                    | 5.52                  |
>     | 10    | 800$\Rightarrow$400 &nbsp;\|   | 0.30             | 3.2                    | 7.96                  |
>     | 10    | 500$\Rightarrow$300 &nbsp;\|   | 0.27             | 0.8                    | 4.85                  |
>     | 5      | 700$\Rightarrow$500 &nbsp;\|   | 0.29             | 2.5                    | 3.32                  |
>     | 20    | 700$\Rightarrow$500 &nbsp;\|   | 0.30             | 3.6                    | 8.83                  |

---

> > ### Author Response · Authors · 2023-11-19
> >
> > Here is the response to the remaining question:
> > - **Q3:** According to Table 1, the proposed method is inferior to SD inpainting on both performance and efficiency. The only superiority of the proposed method is training free. However, since it needs cyclical diffusion & denoising, its inference cost is higher than SD inpainting. The superiority may be weakened.
> > - **A3:**
> >     - Thanks for the comments.
> >     - First, SD inpainting could not handle the cases where the textual condition conflict with the visual condition, both theoretically and empirically. Differently, as provided in the response to Q1 above, our model has the flexibility of fitting in different levels of changes applied to the visual condition, significantly broadening its range of applications in real-world scenarios.
> >     - Second, our quality is far better than SD inpainting, as evidenced by the human study evaluation (Table 2 of the original manuscript). Our model won 69.7% of "best condition consistency'' votes and 51.9% of "best general fidelity'' votes among five models, while SD inpainting only has 6.3% and 2.1% votes, respectively. This clearly shows our model performance is better than SD inpainting. More qualitative results can be seen in Fig. 12 of the appendix.
> >     - Lastly, the cyclical diffusion and denoising are only conducted in part of the diffusion denoising process (20% of the whole process). In addition, since there is large redudant in the cyclical process, we can apply skip steps strategy of DDIM to speed up the cyclic "disturb'' and "construct'' process by 100x.  Thus the overall cost of our model is only slightly higher than the original SD inpainting (6s versus 5s).

---

### Official Review · Reviewer_SshV · 2023-11-04

**Soundness:** 4 excellent
**Presentation:** 4 excellent
**Contribution:** 3 good
**Rating:** 8
**Confidence:** 4

**Summary:**

This manuscript investigates the diffusion-in-diffusion processes
aiming to enable effective both pixel-level and semantic-level
visual conditioning. A cyclic one-way diffusion
method is proposed. The cyclic method starts with an image and builds
the entire scene according to the text information, given a pre-trained
frozen diffusion model.

Extensive experiments are provided for various applications.
Experimental results, included human evaluation are provided.
The experiments and evaluations demonstrate that the proposed method
can generate images with high fidelity to both semantic-text
and pixel-visual conditions.

**Strengths:**

- The manuscript proposes a diffusion-in-diffusion process which is able to enable effective both pixel-level and semantic-level visual conditioning.
- Extensive experiment results are provided for various applications.
-The results indicate that the proposed method can generate images with high fidelity to both semantic-text and pixel-visual conditions.

**Weaknesses:**

-

**Questions:**

What is the level of changes that may occur in the region on the seed image following the cyclic one-way diffusion?
According to the results presented some changes occur sometimes in the region of the seed image from the generated image, but not in
other cases presented.

---

> ### Author Response · Authors · 2023-11-19
>
> Thank you for appreciating our contributions and providing valuable feedback! Please see our response below:
>
> **Q1:** What is the level of changes that may occur in the region on the seed image following the cyclic one-way diffusion? According to the results presented some changes occur sometimes in the region of the seed image from the generated image, but not in other cases presented.
>
> **A1:** The level of changes that may occur within the seed image depends on the discrepancy between textual and visual conditions. In most cases, the seed images can be well preserved given no explicit conflict between the visual-text conditioning pairs. At the same time, appropriate levels of text-guided changes can also be well reflected in the case of attribute editting, style transfer, and cross-domain transformation (as shown in Fig. 7 of the new supplementary file).

---

### Author Response · Authors · 2023-11-19
**Overall Responses to All Reviwers**

We extend our gratitude to the reviewers for their meticulous and insightful feedback, and appreciate the acknowledgment from R-SshV on our effective method, extensive experiment, and various high-quality results; R-S1EF on the clarity of our paper and the elegance and straightforward of our proposed method; R-4D12 on the innovative and practicality of our method and realistic results; R-45wn on the novelty, broad applicability and good performance of our method.

Meanwhile, we would like to emphasize our approach also delivers a significant message to the community: achieving learning-free control over generative output can be more efficient than existing learning-based and step-wise paradigms. Currently, existing methods often incorporate conditional guidance in a step-wise manner by introducing conditional information at each diffusion step throughout the entire denoising process. However, we demonstrate that the above step-wise conditioning is unnecessary to achieve fine-grained control of the generative output. Instead, operating on a critical range is sufficient (20% of the entire steps in this case).

We submit a revised manuscript based on every reviewer's feedback, with major changes highlighted in blue, specifically including more application settings (e.g., whole image generation, cross-domain transformation), more ablation study of hyper-parameters, failure cases analysis, and pseudo code.

---

### Meta-Review · Area_Chair_7EF3 · 2023-12-05

**Metareview:**

The authors propose "cyclic one-way diffusion" which throughout a diffusion process repeatedly replaces the diffusion output with the original visual condition and then encourage the diffusion to operate from the condition and not towards the condition. This is done by repeatedly doing a series step where the condition is continually replaced (similar to restart sampling). They show that this technique allows for adherence to the visual condition.

The main weakness of the paper is the writing. The definition of "diffusion phrases" into, Chaos, Semantic Formation, and Quality Boosting is unscientific and not backed by any evidence or visuals. The description of the "Cylic One-Way Diffusion" is pretty confusing with the usage of the arbitrary words: "disturb" and "reconstruct" as opposed to standard diffusion terminology.

**Justification For Why Not Higher Score:**

The writing prevents this from being more than a borderline accept.

**Justification For Why Not Lower Score:**

I think this paper has a technical contribution that's interesting. With that said, I would not object to having this paper be rejected and resubmitted somewhere else with the writing improved.

---

### Decision · Program_Chairs · 2024-01-16

Accept (poster)